# High Mitochondrial Protein Expression as a Potential Predictor of Relapse Risk in Acute Myeloid Leukemia Patients with the Monocytic FAB Subtypes M4 and M5

**DOI:** 10.3390/cancers16010008

**Published:** 2023-12-19

**Authors:** Frode Selheim, Elise Aasebø, Øystein Bruserud, Maria Hernandez-Valladares

**Affiliations:** 1Proteomics Unit of University of Bergen (PROBE), University of Bergen, Jonas Lies vei 91, 5009 Bergen, Norway; 2Acute Leukemia Research Group, Department of Clinical Science, University of Bergen, Jonas Lies vei 91, 5009 Bergen, Norway; elise.aasebo@uib.no (E.A.); oysteinbruserud@yahoo.no (Ø.B.); 3Section for Hematology, Department of Medicine, Haukeland University Hospital, 5009 Bergen, Norway; 4Department of Physical Chemistry, Institute of Biotechnology, Excellence Unit in Chemistry Applied to Biomedicine and Environment, School of Sciences, University of Granada, Campus Fuentenueva s/n, 18071 Granada, Spain; 5Instituto de Investigación Biosanitaria ibs.GRANADA, 18012 Granada, Spain

**Keywords:** acute myeloid leukemia, mass spectrometry, proteomic, phosphoproteomic, relapse, mitochondria, FAB subtypes

## Abstract

**Simple Summary:**

Proteins serve as the primary regulators of cellular functions, and the development of new drugs largely focuses on target proteins that play crucial roles in specific diseases. Quantitative proteomics has emerged as a promising analytic technique, offering the potential to identify disease-related proteins and develop novel biomarkers linked to prognosis and subclassification of specific cancer types. The aim of this study was to examine the potential of mass spectrometry-based proteomic profiling in identifying distinct protein expression and phosphorylation patterns that may help predict relapse risk in acute myeloid leukemia (AML) patients with different French-American-British (FAB) subtypes. Our approach exposed differential protein expression and regulation of phosphorylated sites among various FAB subtypes. Moreover, the presence of high levels of mitochondrial proteins at diagnosis predicts an unfavorable prognosis with a high relapse rate for patients who exhibit the FAB M4/M5 subtype.

**Abstract:**

AML is a highly aggressive and heterogeneous form of hematological cancer. Proteomics-based stratification of patients into more refined subgroups may contribute to a more precise characterization of the patient-derived AML cells. Here, we reanalyzed liquid chromatography-tandem mass spectrometry (LC-MS/MS) generated proteomic and phosphoproteomic data from 26 FAB-M4/M5 patients. The patients achieved complete hematological remission after induction therapy. Twelve of them later developed chemoresistant relapse (RELAPSE), and 14 patients were relapse-free (REL_FREE) long-term survivors. We considered not only the RELAPSE and REL_FREE characteristics but also integrated the French-American-British (FAB) classification, along with considering the presence of nucleophosmin 1 (*NPM1*) mutation and cytogenetically normal AML. We found a significant number of differentially enriched proteins (911) and phosphoproteins (257) between the various FAB subtypes in RELAPSE patients. Patients with the myeloblastic M1/M2 subtype showed higher levels of RNA processing-related routes and lower levels of signaling related to terms like translation and degranulation when compared with the M4/M5 subtype. Moreover, we found that a high abundance of proteins associated with mitochondrial translation and oxidative phosphorylation, particularly observed in the RELAPSE M4/M5 *NPM1* mutated subgroup, distinguishes relapsing from non-relapsing AML patient cells with the FAB subtype M4/M5. Thus, the discovery of subtype-specific biomarkers through proteomic profiling may complement the existing classification system for AML and potentially aid in selecting personalized treatment strategies for individual patients.

## 1. Introduction

Acute myeloid leukemia (AML) is a highly aggressive type of blood cancer that arises from hematopoietic stem or progenitor cells. Its heterogeneity is attributed to several factors, including different mutations, potential cytogenetic abnormalities, changes in gene and protein expression, and disrupted signaling transduction [1,2]. Prior to the identification of genetic and cytogenetic abnormalities, the classification of AML into subtypes relied primarily on the morphological characteristics of the leukemic cells. In the 1970s, the French-American-British (FAB) Cooperative Group proposed a classification system for AML, which divided AML patients into eight FAB subgroups (M0–M7) based on morphological, cytochemical, and maturation characteristics of the leukemic cells [3]. Later, the importance of cytogenetics and molecular genetic features in the stratification of patients into risk groups, such as those with favorable prognosis and high complete remission (CR) rates, as well as intermediate and poor/adverse outcomes, was recognized [4,5]. Based on new knowledge of clinical and genetic abnormalities, the World Health Organization (WHO) and the European Leukemia Network (ELN) have recently updated their risk classification and treatment recommendations [1,2]. At initial AML diagnosis, patients with nucleophosmin 1 (*NPM1*) mutation without Fms-related receptor tyrosine kinase 3-internal tandem duplication (*FLT3*-ITD) are categorized as favorable, whereas mutated *NPM1* along with *FLT3*-ITD are now classified as intermediate risk in the revised ELN risk classification [1]. Additionally, mutated *NPM1* with adverse-risk cytogenetics are now classified as adverse.

In adults, the *NPM1* mutation and morphological signs of differentiation, along with the expression of the CD33 differentiation marker and absence of the CD34 stem cell marker, are more commonly observed in the monocytic FAB-M4/M5 subgroups and less frequently seen in the myeloblastic FAB-M0/M1/M2 subgroups [6,7,8]. Monocytic differentiation is also associated with generally high constitutive cytokine release [9,10], i.e., these cells differ with regard to the communication with neighboring stromal cells in their common bone marrow microenvironment. Gene expression profiling of AML has provided valuable insight into distinct gene expression signatures observed in different patient subgroups characterized by specific genetic and cytogenetic abnormalities [7,11]. These profiles include unique gene expression patterns associated with *NPM1* mutations and also the expression levels of three genes (annexin A3, *ANXA3*; protein S100-A9, *S100A9*, and Wilms tumor 1, *WT1*) that can differentiate between AML FAB subtypes M1 from M2 [12,13]. Similarly, proteomic profiling was conducted to compare differences in protein expression between two subtypes of myeloblastic AML: M1 without maturation and M2 with maturation. The study identified five proteins (ANXA A1; ANXA A3; plastin-2, PLSL; 6-phosphogluconate dehydrogenase, 6PGD; actin cytoplasmatic 2, ACTG) that exhibited differential expression, allowing for the distinction between the two subtypes [14].

Recent advancements in quantitative proteomics, especially those based on liquid chromatography-tandem mass spectrometry (LC-MS/MS), have made it possible to accurately quantify AML-disease-related proteins and phosphorylation sites in a substantial number of patients with different disease characteristics and treatment responses [15,16,17,18,19,20,21,22,23,24,25]. In a previous study, we employed quantitative LC-MS/MS analysis to compare the proteome and phosphoproteome of pretreatment AML cells obtained at the time of diagnosis. Specifically, we focused on two distinct patient subgroups: 15 patients who achieved leukemia-free survival for more than five years and 26 patients who experienced relapse despite undergoing intensive chemotherapy (henceforth REL_FREE and RELAPSE patients, respectively) [26]. The heterogeneity of AML emphasizes the importance of the stratification of patients into disease subgroups. Here, we further categorized the original cohort of 41 AML patients into more refined subgroups, considering not only relapse categories but also incorporating the FAB-M1/M2 and FAB-M4/M5 classification subgroups (i.e., morphological signs of AML cell differentiation), as well as *NPM1* mutation and cytogenetically normal AML. The aim of our present study was thus to identify at the first time of diagnosis the heterogeneity of proteomic and phosphoproteomic AML cell profiles for patients that later develop leukemia relapse after intensive and potentially curative chemotherapy.

## 2. Materials and Methods

### 2.1. Patient and Sample Collection

We here reanalyzed our previously published LC-MS/MS-based proteomic and phosphoproteomic cohort of primary cells from 41 AML patients at the time of diagnosis [26]. These patients represent a consecutive group of relatively young patients who received intensive and potentially curative antileukemic treatment. The study should be regarded as population-based because our department was responsible for the diagnosis and treatment of AML in a defined geographical area during the defined time period. The patients received only intensive induction and consolidation cytotoxic therapy without stem cell transplantation as their initial therapy, and REL_FREE patients after this treatment were classified after observation for at least five years.

Primary AML cells were density gradient-separated from peripheral blood (PB) of untreated patients with blast counts (leukemic cells) exceeding 80% of the circulating leukocytes. The cells were cryopreserved and stored in liquid nitrogen until analyzed. Quantification was performed by combining protein lysates with a heavy-labeled AML-super SILAC (stable isotope labeling by amino acids in cell culture) mixture [27]. Detailed methods and patient information on FAB type, cytogenetic, and mutational analysis from the time of first diagnosis are comprehensively given in our previously described cohort of 41 AML patients [26,28]. The criteria for FAB classification of patients have been described in detail previously [29]; this system is regarded as a standardized and well-described system to characterize and classify AML patients with regard to the differentiation status of their leukemic cells [30,31]. All raw data and MaxQuant output files can be found in the ProteomeXchange consortium with the dataset identifier PXD014997.

### 2.2. Data Analysis

Patients were grouped after clinical progression as RELAPSE and REL_FREE if they had not relapsed after a five-years observation time from the initial induction chemotherapy/consolidation therapy. The Perseus 2.0.7.0 bioinformatics platform was used for functional and statistical analysis of the proteomics and phosphoproteomics data [32]. SILAC ratios were inverted, and log2 transformed. Categorical annotation rows were used for stratification of patients into disease subgroups based on FAB classification, mutated *NPM1* and cytogenetic status (Table 1). The generated AML subgroups included eight RELAPSE M1/M2 (REL_M1/2_all), 12 RELAPSE M4/M5 (REL_M4/5_all), and 14 REL_FREE M4/M5 (REL_F_M4/5_all). M0 patients were left out as there were no M0 REL_FREE patients, and four out of the five M0 RELAPSE patients did not show *NPM1* mutations.

Within the REL_M1/2_all subgroup, only two patients showed a 4-base pair insertion (Ins) mutation in *NPM1*, and five patients had a normal cytogenetic status. This subgroup was not further stratified for analysis. Within the M4/M5 patients, five RELAPSE M4/M5 and eight REL_FREE M4/M5 showed the Ins *NPM1* mutation (REL_M4/5_mut and REL_F_M4/5_mut, respectively); and seven RELAPSE M4/M5 and nine REL_FREE M4/M5 displayed a normal cytogenetic status 46, XY, or XX (REL_M4/5_CN and REL_F_M4/5_CN, respectively). These subgroups were normalized by using width adjustment. Proteins and phosphosites (localization probability > 0.75) with a minimum of three individual SILAC ratios for each patient subgroup were selected for statistical analysis. An ANOVA multiple sample test was performed with a threshold *p*-value < 0.05 to test for significant differences between means for the proteins and phosphosites between the subgroups. Hierarchical clustering of significantly differential proteins with ANOVA was done using the Euclidean function and complete linkage. A post hoc Turkey’s honest significance difference (HSD) test with FDR < 0.05 was performed on the ANOVA significant different pairs of protein and phosphosites. Reactome pathway, Gene Ontology (GO), and KEGG pathway enrichment analyses were obtained with the Enrichr gene set search engine [33,34,35]. Protein–protein interaction (PPI) network analysis was performed with STRING database version 11.5 [36]. Networks were visualized using the Cytoscape platform v3.10.0 [37]. The ClusterONE plugin was used to identify protein groups of high cohesiveness [38]. Phosphosite motif analysis was performed with the web-based WebLogo application [39]. Venn diagrams were generated by BioVenn (https://www.biovenn.nl/) [40].

## 3. Results

### 3.1. Patient Characteristics and Patient Subclassification as a Basis for Bioinformatical Analyses

The differentiation block is a fundamental biological characteristic of the AML cells, although the localization/degree of differentiation block varies between patients. In the present study, we classified our patients based on morphological signs of differentiation as described in the original FAB classification that was referred to in the previous WHO 2016 classification [5]. Our study included only patients with non-APL (acute promyelocytic leukemia) variants of AML. All included patients were relatively young and fit for intensive and potentially curative conventional AML therapy, and they all achieved complete hematological remission after the initial induction therapy. None of the patients received treatment with BCL-2 family member inhibitors.

The recently published WHO 2022 classification of AML is based on patient history (previous chemotherapy or hematological disease), karyotype, and molecular genetic analyses, including the molecular abnormalities ASXL1, BCOR, EZH2, RUNX1, SF3B1, SRSF2, STAG2, U2AF1, and ZRSR2 referred to as associated with myelodysplastic syndrome associated [41]. All our patients had >20% AML blasts in the bone marrow, none of the patients had evidence of germline predispositions, and previous hematological diseases fulfilled the diagnostic criteria as outlined in the WHO 2022 classification [41]. Detailed patient information is given in Appendix A. These tables present more in detail the clinical and biological characteristics (age, gender, disease history, karyotyping and molecular genetic analyses) of each individual patient included in the present study, and the patients are then classified into three groups based on the differentiation block: FAB-M1/M2 with later relapse (Appendix A), FAB-M4/M5 monocytic long-term survivors, and FAB-M4/M5 monocytic with later relapse (Appendix A). Due to the low number of patients, FAB-M1/M2 long-term survivors are not included in this study (Table 1). Although extensive molecular genetic analyses are not available for all patients, it can be seen from this table that each of these three groups was very heterogeneous with regard to clinical and biological characteristics, i.e., our classification based on the differentiation block goes across both the recent WHO 2022 [41] and the most recent ELN [1] subclassification of AML patients.

Given the high number of possible individual comparisons among the patient subgroups (Table 1), proteomic data results will be presented as a Venn diagram of the regulated proteins from different subgroup comparisons. Enrichment of Reactome pathways and PPI of overlapped and subgroup-specific regulated proteins will be shown as well. Phosphoproteomic data results will be additionally presented as sequence logos of the surrounding amino acids of the differentially regulated phosphorylation sites obtained from overlapped and subgroup-specific phosphosites.

### 3.2. Distinct Protein Expression and Site Phosphorylation Patterns in RELAPSE Patients for AML FAB Subtypes M1/M2 and M4/M5

The heterogeneity of AML stresses the importance of categorizing patients into disease subgroups based on various considerations such as cytogenetic, genetic mutations, protein expression, and aberrant post-translational modification (PTM) patterns. Based on information about clinical progression and pathological processes, we recategorized our original cohort with 41 AML patients [26] into more defined subgroups based on FAB, cytogenetic, and *NPM1* mutation parameters (Table 1). We quantified a total of 6781 proteins and 12,309 class I protein phosphorylation sites. Among these, 4601 proteins and 3148 phosphosites had at least three valid SILAC ratios in each patient subgroup (Appendix A). By comparing the different AML disease subgroups for patients that relapse after chemotherapy, we found a substantial number of proteins (911) and phosphosites (257) that exhibited statistically significant (ANOVA, post hoc Turkey’s HSD with FDR < 0.05) differences between the FAB classes (RELAPSE M1/M2 vs. RELAPSE M4/M5 subgroups, Table 1): 162 proteins and 32 phosphosites were upregulated, and 268 proteins and 94 phosphosites were downregulated for the comparison REL_M1/2_all vs. REL_M4/5_all; 34 proteins and 24 phosphosites were upregulated, and 283 proteins and 86 phosphosites were downregulated for REL_M1/2_all vs. REL_M4/5_mut; and 131 proteins and 44 phosphosites were upregulated, and 382 proteins and 89 phosphosites were downregulated for REL_M1/2_all vs. REL_M4/5_CN (Appendix A).

Seventy-eight regulated proteins were identified in the REL_M1/2_all comparisons against the different REL_M4/5 subgroups (Figure 1a). Through the utilization of Reactome pathways and PPI network analyses, we observed distinct patterns of protein expression linked to the AML FAB subtypes M1/M2 and M4/M5. The basal transcription machinery, such as RNA processing and RNA polymerase transcription, were significantly enriched in RELAPSE patients with the myeloblastic subtype M1/M2 (Figure 1b,c), whereas terms like translation, neutrophil degranulation, and intracellular protein/vesicle transport were more abundant terms for RELAPSE patients with the monocytic subtypes M4/M5 (Figure 2b,c). Additionally, we found that hematopoietic stem cell differentiation was enriched for the M4/M5 subtype (Figure 2c). We observed a high number of regulated proteins (166) that were detected in the different REL_M4/5 comparisons against REL_M1/2_all (Figure 2a).

Thirteen differentially regulated phosphorylation sites were identified in the REL_M1/2_all comparisons against the different REL_M4/5 subgroups (Figure 3a). By conducting GO and KEGG pathway analyses on the differently regulated phosphorylation sites, we confirmed the enrichment of basal transcription machinery, as well as RNA and DNA binding processes, in relapse patients with M1/M2 subtype compared with the M4/M5 subtypes (Figure 3b). Forty-two differentially regulated phosphorylation sites were identified in the different REL_M4/5 comparisons against REL_M1/2_all (Figure 4a). The relapse M4/M5 patient subgroups exhibited higher site-specific phosphorylation on proteins linked to RNA binding and high translational activity when compared with the M1/M2 subgroup (Figure 4b). PPI network analysis confirmed the higher phosphorylation of translational proteins in the REL_M4/5_all, REL_ M4/5_mut and REL_M4/5_CN patient groups (Figure 4c). Additionally, the REL_M4/5_all and REL_M4/5_CN subgroups showed increased phosphorylation of proteins involved in DNA damage response and protein synthesis.

To identify the potential kinases responsible for phosphorylating the differently regulated phosphosites, we conducted WebLogo substrate motif analysis. As illustrated in Figure 3c, we observed a prevalent pSP followed by a pSXXE motif in REL_M1/2_all compared with all RELAPSE M4/M5 subtypes, suggesting higher activity of extracellular signal-regulated kinases (ERK1/2) and casein kinase 2 (CSNK2) in the M1/M2 subtype. The pSP motif for ERK1/2 and RXXpS motif for protein kinase A (PRKA) and C (PRKC) were prominent in REL_M1/2_all compared with REL_M4/5_all and REL_M4/5_CN. Multiple kinase substrate motifs, i.e., pSP, pSXXE/D, and RXXpS, were found in the REL_M1/2_all vs. REL_M4/5_CN comparison.

The pSP and RXXpS substrate motifs were found in all the REL_M4/M5 subgroups compared with REL_M1/2, indicating increased activity of ERK1/2 and PRKA/PRKC in AML patients with the subtype M4/M5 (Figure 4d). However, the acidic amino acids in close proximity to the differentially regulated phosphorylation sites of the CSNK2 substrate motif were found in the overlapping sequences flanking the regulated phosphosites from the REL_M4/5_all vs. REL_M1/2_all and REL_M4/5_CN vs. REL_M1/2_all and in the separated REL_M4/5_CN vs. REL_M1/2_all and REL_M4/5_mut vs. REL_M1/2_all comparisons.

### 3.3. High Mitochondrial Protein Expression Splits Relapsing from Non-Relapsing AML Patients with the FAB Subtypes M4/M5

We conducted a comparison between the proteome and phosphoproteome profiles of patients with the monocytic FAB-M4/M5 subtypes who later experienced relapse to those who did not. Among the 4601 proteins and 3148 phosphosites which had at least three valid SILAC ratios in each patient subgroup, we found a substantial number of proteins (850) and phosphosites (294) that exhibited statistically significant (ANOVA, post hoc Turkey’s HSD with FDR < 0.05) differences between the relapse status (RELAPSE M4/M5 vs. REL_FREE M4/M5 subgroups; Table 1, Appendix A).

For the FAB-M4/M5 group with *NPM1* mutation, we quantified 193 differently expressed proteins and 34 differentially regulated phosphorylation sites for RELAPSE and REL_FREE patient comparisons (ANOVA, post hoc Turkey’s HSD with FDR < 0.05). Among these, 145 proteins and 17 phosphosites were more abundant, while 48 proteins and 17 phosphosites were less abundant for the REL_M4/5_mut vs. REL_F_M4/5_mut comparison (Appendix A). The number of regulated proteins and phosphosites for the FAB-M4/M5 subgroup with normal cytogenetics or without any other stratification for RELAPSE compared with REL_FREE patients was low (Figure 5a). Through Reactome pathways enrichment analysis, we discovered that terms like mitochondrial translation were significantly enriched in the REL_M4/5_mut subgroup (Figure 5b). Importantly, proteins associated with mitochondrial translational activity were also more abundant in REL_M4/5_all and REL_M4/5_CN when compared with the corresponding REL_FREE subgroups. Moreover, significant enrichment of PPI networks required for mitochondrial translation, electron transport, and ATP synthesis was found in RELAPSE patients with FAB subtypes M4/M5 (Figure 5c, Appendix A). The largest network cluster consisted of 37 mitochondrial translational elongation proteins observed in the REL_ M4/5_mut subgroup. The mitochondrial protein interaction networks reflected differences in the mitochondrial energy metabolism and included 11 proteins that are important for the mitochondrial ATP synthase (the final step in the ATP-generating electron chain) as well as proteins important for Complex I (NDUFB3, NDUFC2, NDUFST, NDUFV2), Complex III (UQCRC2, UQCRQ), and cytochrome c oxidase (COX15, COX5B) of the mitochondrial electron chain [42,43,44]. Additionally, this *NPM1*-mutated REL_M4/5 subgroup exhibited functional PPI clusters for transcription elongation from RNA polymerase II promotor. Hierarchical clustering analyses showed that mitochondrial ribosomal proteins of the translation elongation network and ATP synthases appeared to be better discriminators than electron transport proteins (NADH dehydrogenases) between REL_M4/5 and REL_F_M4/5 patients (Appendix A).

On the other hand, we did not find any regulated protein from the REL_F_M4/5_all vs. REL_M4/5_all comparison (Figure 6a, Appendix A), whereas REL_F_M4/5_mut and REL_F_M4/5_CN subgroups were generally more prevalent in terms related to metabolism of nucleotides when compared with the RELAPSE counterparts. Moreover, endosomal sorting, Golgi-to-ER retrograde traffic, and regulation of actin cytoskeleton were also enriched in the REL_F_M4/5_mut subgroup (Figure 6b,c).

Regarding the phosphoproteome, a few differentially regulated phosphorylation sites were identified from the different subgroup comparisons ((Figure 7a and Figure 8a, Appendix A). However, we observed higher site-specific phosphorylation on proteins associated with transcription and translation, such as 90S preribosome, mRNA splicing, and poly (A)+ mRNA export in the REL_M4/5_mut subgroup compared with the REL_F_M4/5_mut (Figure 7b). Additionally, GO terms like chromatin organization and DNA binding were significantly enriched in the REL_M4/5_mut subgroup. Through kinase substrate motif analyses, we identified a prevailing pSP motif for ERK1/2 seconded by the pSXXD/E motif for CSK2 in the REL_M4/5_mut subgroup (Figure 7c), while the RXXpS motif for PRKA/PRKC was most prominent in the REL_F_M4/5_mut subgroup (Figure 8c). Interestingly, this subgroup exhibited significantly enriched site-specific phosphorylation on proteins associated with glycolysis and gluconeogenesis (Figure 8b). Moreover, positive regulation of autophagy of mitochondrion was also overrepresented.

### 3.4. High BH3 Interacting Domain Death Agonist (BID) Protein Expression in Non-Relapsing AML Patients with the FAB Subtypes M4/M5

The BCL-2 family consists of antiapoptotic and proapoptotic proteins that control the permeabilization of the mitochondrial outer membrane.

We compared the BCL-2 protein expression of patients with the monocytic FAB-M4/M5 subtypes who later experienced relapse to those who did not. The proapoptotic BID protein showed up to four times higher abundance in REL_F_M4/5_mut cells compared with REL_M4/5_mut (ANOVA, *p* < 0.05). Additionally, it was twice as abundant in REL_F_M4/5_all and REL_F_M4/5_CN compared with RELAPSE patients (REL_M4/5_all and REL_M4/5_CN, Appendix A).

Interestingly, tBID alone has recently been reported to be sufficient to trigger permeabilization of the mitochondrial membrane and induce apoptosis [45].

No significant differential expressions were observed for other members of the BCL-2 family, including the antiapoptotic/pro-survival protein BCL-2 and the proapoptotic BAX protein.

## 4. Discussion

The FAB classification system provides a standardized and well-described system to characterize and classify AML patients with regard to the differentiation status of their leukemic cells [30,31]. The FAB classification has been replaced by the WHO and ELN classification and is no longer considered to have a prognostic role when the mutation status of *NPM1* and CCAAT/enhancer-binding protein alpha (*CEPBA*) is known [46].

Even though the differentiation block is regarded as a fundamental characteristic of the AML cells [41], as stated above, the differentiation-based FAB classification (i.e., degree of differentiation block) has a very limited prognostic impact for relatively young and fit leukemia patients receiving conventional intensive and potentially curative antileukemic therapy. Two previous studies have investigated the possible impact of FAB/differentiation on survival for AML patients receiving allogeneic stem cell transplantation. A small early study included 39 patients (median age 14 years) transplanted in the period from November 1976 to July 1983. These authors described an adverse prognostic impact of high peripheral blood leukocyte counts at the time of diagnosis (i.e., ≥ 20 × 10^9^/L; *p* = 0.001) and monocyte morphology (i.e., FAB M4/M5, *p* = 0.05) [47]. It should also be mentioned that most patients in the FAB-M4/M5 groups died from relapse, whereas most of the other patients died in remission, but the numbers of patients are low, and a reliable statistical comparison is therefore not possible. Another study was based on 1690 patients transplanted in first complete remission [48]. The patients were classified as having AML not otherwise specified according to the 2016 WHO classification, and the authors described an association between FAB M6/M7 and adverse prognosis, i.e., increased nonrelapse mortality. Finally, the possible association between differentiation and survival after allogeneic stem cell transplantation may not only reflect an association between differentiation and susceptibility to antileukemic treatment; posttransplant survival is possibly also influenced by the immunomodulatory effects (i.e., inhibition of antileukemic immune reactivity) through the expression of immune checkpoint ligands by the AML cells [49].

In contrast to the observations in patients receiving conventional intensive chemotherapy (see above), several studies of new targeted therapies suggest that AML cell differentiation is important for responsiveness to these therapies and/or differentiation is induced as a part of the response to the treatment [49]. First, pretherapy signs are monocytoid [30,31,50] or erythroid [51]. AML cell differentiation is associated with decreased responsiveness to the BCL-2 inhibitor venetoclax. Second, differentiation induction is a part of the antileukemic effect for several new/targeted anti-AML therapies, including FLT3 [52,53], isocitrate dehydrogenase (IDH) [54], bromodomain [55], lysine demethylase 1 (LSD1) [56], DOT1-like histone H3K79 methyltransferase (DOT1L) [57], exportin (XPOT) [58], menin (MEN) [59], and pyrimidine metabolism [60] inhibitors. Third, even clinical differentiation syndrome can be observed in responders to several of these targeted therapies [49]. Finally, differentiation induction can be required for the synergistic effect of combining various new targeted therapies, e.g., the combination of IDH and BCL-2 inhibitors [61]. Altogether, these observations illustrate the importance of leukemic cell differentiation for AML cell biology and responsiveness to new targeted therapies, and our present results suggest that mechanisms involved in AML cell differentiation/differentiation block can be relevant for and/or contribute to the risk of AML relapse even after conventional antileukemic treatment.

Even though experimental studies suggest that FAB subtypes differ with regard to the antileukemic effects of daunorubicin and cytarabine (i.e., two drugs commonly combined in the initial induction treatment of AML) [62], the overall results from the clinical studies discussed above suggest that FAB classification/AML cell differentiation has a limited prognostic impact for patients receiving conventional antileukemic treatment. However, the studies of AML, in general, do not exclude a possible impact of differentiation for patient subsets and/or other types of antileukemic treatment.

Recently, Wojtuszkiewicz et al. found that there is maturation state-specific differential splicing of genes associated with cell cycle control and DNA damage in *FLT3*-ITD and *NPM1*-mutated AML blasts. Intriguingly, the number of genes that displayed differential splicing was significantly higher in the FAB M4 subtype, with a total of 1438 splicing events, compared with the FAB M1 and M2 subtypes, each with about 200 splicing events [63].

The FAB-M4/M5 subset of AML patients is a heterogeneous group with regard to genetic abnormalities and includes mutations associated with both adverse and favorable prognoses for patients receiving intensive treatment based on conventional cytotoxic drugs [47,64,65,66,67]. The present study demonstrates that distinct patterns of protein expression and phosphorylation, as well as signaling pathways, are associated with each of the different FAB subtypes, specifically M1/M2 and M4/M5. In total, we found 911 proteins and 257 phosphosites that exhibited differential regulation when comparing the RELAPSE M1/M2 subtype to all the different subtypes of RELAPSE M4/M5. In RELAPSE patients with the myeloblastic subtype M1/M2, RNA-related processes like transcription and splicing were significantly increased. On the other hand, in RELAPSE patients with the more differentiated monocytic subtype M4/M5, there was a higher prevalence of signaling pathways involved in translation and degranulation. Additionally, the kinase-substrate analysis demonstrated enrichment of ERK1/2 and CSK2 kinases in the RELAPSE M1/M2 subgroup, whereas PKRA and PKRC kinases exhibited higher activity in the M4/M5 subgroups. In a study by Kornblau et al., a reverse-phase protein array (RPPA) was used to distinguish between different AML FAB subtypes in 256 patients. They identified 24 proteins that showed differential expression among the 51 assayed proteins, effectively separating the myeloblastic subtype M1/M2 from the monocytic subtype M4/M5 [68]. Thus, the unique expression patterns of proteins and phosphoproteins among the different FAB subtypes, as identified in both current and previous studies [68,69], point out the importance of incorporating FAB classification into proteomic and phosphoproteomic studies. A recent study suggested that malignant hematological cells share biological characteristics with their normal counterparts [70]; this seems to also be true for monocytic AML cells (i.e., FAB-M4/M5 subclassification that shows high levels of constitutive release of several cytokines/chemokines as well as other soluble mediators) [10,70,71,72]. Normal macrophages seem to be reprogrammed by AML cells and thereby support leukemogenesis/chemosensitivity of the leukemic cells through their release of chemokines/cytokines [73]; when these mediators are released by the AML cells, they may become a part of the intrinsic mechanisms for chemoresistance/disease progression caused by autocrine mechanisms and associated with differentiation in primary human AML cells.

We have previously reported that phosphoproteins and proteins linked to ribosome biogenesis and rRNA processing exhibit higher abundance in AML cells obtained from patients who experience relapse after intensive chemotherapy compared with patients who achieve leukemia-free survival (>5 years) [26]. In this study, we performed further stratification of this initial cohort by incorporating FAB classification and subsets with *NPM1* mutation and normal cytogenetics as additional selection criteria. We found 850 proteins and 294 phosphosites that exhibited statistically significant differences between the relapse status, RELAPSE M4/M5 vs. REL_FREE M4/M5 subgroups. In addition to confirming previous findings of term enrichment related to transcription and high CSK2 kinase activity, our current study also revealed a significant enrichment of proteins associated with mitochondrial translation and oxidative phosphorylation, particularly in the REL_M4/5_mut subgroup, as well as in all M4/M5 subgroups of patients who experienced relapse. The proteins found to be enriched include various mammalian mitochondrial ribosomal proteins (MRPL, MRPS), the mitochondrial intermediate peptidase (MIPEP) involved in the processing of oxidative phosphorylation-related proteins within the mitochondria [74], and NADH:ubiquinone oxidoreductase complex assembly factor 2 (NDUFAF2), a constituent of the NADH:ubiquinone oxidoreductase (complex I). Complex I is responsible for catalyzing the transfer of electrons from NADH to ubiquinone, which is the initial step in the mitochondrial respiratory chain [75]. Moreover, individual proteomic profiling showed that mitochondrial ribosomal proteins and ATP synthases could be used as relapse predictors in FAB M4/M5-classified patients with the *NPM1* Ins mutations. However, these findings require further validation with external cohorts. Another intriguing observation in the present study is the significant enrichment of site-specific phosphorylation sites associated with glycolysis and autophagy of the mitochondrion and the involvement of PRKA/PRKC kinases in the REL_F_M4/5_mut subgroup, as compared to the REL_ M4/5_mut patients.

Patients with AML FAB-M4/M5 are heterogeneous with regard to their genetic abnormalities and include abnormalities with both favorable and adverse prognostic impact [65]. In our present study, we observed that FAB-M4/M5 patients with later relapse had a proteomic profile that differed both from FAB-M4/M5 patients without relapse and from other relapse patients (i.e., FAB-M1/M2 patients). Despite their genetic heterogeneity, relapsed FAB-M4/M5 patients had common proteomic differences with regard to mitochondrial function when compared to other FAB-M4/M5 patients. Even though previous studies have failed to demonstrate a prognostic impact of FAB-M4/M5 in AML patients receiving intensive and potentially curative cytotoxic therapy [46], our present study suggests that the molecular mechanisms behind relapse differ between patients and for certain subsets (at least partly) depend on the AML cell differentiation.

Mitochondria are important regulators of both cellular metabolism and survival; these two regulatory systems are characterized by similar compartmentalization but also by molecular crosstalk/interactions, and the apoptotic machinery (including apoptosis regulator BCL-2) is involved in the regulation of mitochondrial metabolism [76]. Our observation that the mitochondrial function/metabolism at the first time of diagnosis differs between AML-FAB-M4/M5 patients with and without later relapse is also consistent with other observations suggesting that mitochondrial function/energy metabolism is important for susceptibility to antileukemic therapy. First, monocytic differentiation reflected by the FAB classification is associated with resistance to venetoclax-based (i.e., a BCL-2 inhibitor), and this resistance seems to be due to decreased functional importance of BCL-2 and thereby altered regulation of apoptosis and mitochondrial energy metabolism in monocytic AML cells, including leukemic stem cells [30,31,50,77]. Second, differential expression of mitochondria-related genes is important for chemoresistance and seems to have an independent prognostic impact in AML [78]. Third, a subset of AML patients show mutations in genes that encode proteins in the electron transport complexes (Complex I/III/IV, ATP synthase), and mutations in the mitochondrial NADH dehydrogenase subunit 4 (a component of Complex I) seem to have a prognostic impact in adult AML [79,80]. Finally, monocytic differentiation is associated with response to BET (bromodomain and extraterminal domain protein family) inhibitors [55]. Altogether, these studies show that monocytic AML cell differentiation together with mitochondrial functions are important for the responsiveness to various forms of antileukemic strategies, and this is also the reason why oxidative phosphorylation is regarded as a possible therapeutic target in cancer therapy [43]. Moreover, previous studies with venetoclax have observed variations between antiapoptotic members depending on differentiation: (i) AML cells with erythroid or megakaryocytic differentiation depend on the antiapoptotic protein B-cell lymphoma (BCL)-XL, rather than BCL-2 [51], (ii) resistant monocytic AML has a distinct transcriptomic profile, loses expression of venetoclax target BCL-2, and relies on induced myeloid leukemia cell differentiation protein Mcl-1 (MCL1) to mediate oxidative phosphorylation and survival [31], (iii) and other forms rely on BCL-2 [31]. This study then indicates an association between an apoptotic agonist and susceptibility to conventional antileukemic therapy. This further illustrates that the prognostic impact of various individual BCL-2 family members (i.e., responsiveness to antileukemic therapy) depends both on the differentiation status (MCL1 for venetoclax in M4/M5) and the type of antileukemic therapy (BID for M4/M5 receiving conventional therapy). This is in accordance with how we think about prognostic parameters in cancer therapy in general. It is also easy to accept that the effect of a treatment targeting a specific cellular mechanism will depend on the cellular biological/molecular context that influences this mechanism.

Our present study suggests that the importance of altered mitochondrial function/metabolism for the development of AML relapse after intensive cytotoxic treatment differs between patients and is of particular importance in AML cells showing monocytic differentiation. We were the first to demonstrate that chemoresistant relapsed AML cells have transitioned to a state characterized by higher expression of mitochondrial proteins in adults [81]. In line with our findings, Stratman et al. recently published a comprehensive proteogenomic study reporting that the proteome at relapse is enriched with mitochondrial ribosomal proteins and subunits of the mitochondrial respiratory chain complex, not only in adults but also in children [24]. Interestingly, recent evidence suggests the existence of a proteomic subtype called Mito-AML, characterized by elevated expression of mitochondrial proteins and associated with a poor outcome. Moreover, Mito-AML cells exhibit a strong reliance on complex I-dependent mitochondrial respiration, which can be targeted by drugs like venetoclax [21]. Thus, both current and previous proteomics studies indicate that increased mitochondrial translational activity and oxidative phosphorylation are associated with poor prognosis, including higher relapse rates and reduced overall survival.

## 5. Conclusions

This and recently published proteomics studies demonstrate that high mitochondrial protein abundance and respiration are associated with higher relapse rates and reduced overall survival [21,24,81]. The identification of distinct protein expression and phosphorylation profiles for each AML subtype, as observed in the present study for the FAB subtypes M1/M2 and M4/M5, holds promise for the discovery of subtype-specific biomarkers. These biomarkers could serve as predictors of prognosis and potential targets for personalized therapies. In particular, the high expression of mitochondrial ribosomal proteins and associated respiratory chain complexes emerges as a reliable predictor of a high relapse risk in AML patients with M4 and M5 subtypes.

## Figures and Tables

**Figure 1 cancers-16-00008-f001:**
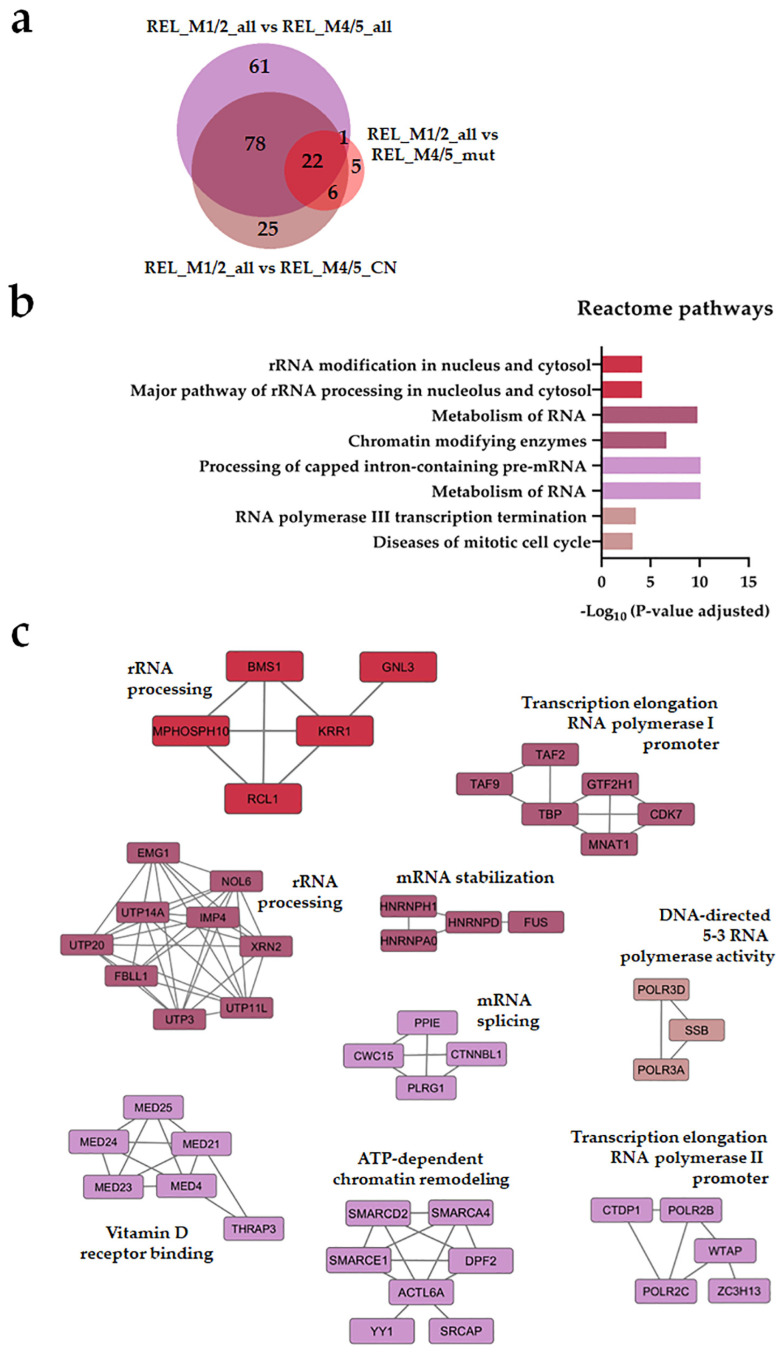
The AML cell proteome shows increased abundance of rRNA metabolism and modification proteins for RELAPSE patients with FAB classification M1 and M2. (**a**) Venn diagram of regulated proteins (ANOVA, post hoc Turkey’s HSD with FDR < 0.05) obtained from REL_M1/2_all vs. REL_M4/5_all, REL_M1/2_all vs. REL_M4/5_mut, and REL_M1/2_all vs. REL_M4/5_CN comparisons. (**b**) Reactome pathways and (**c**) protein–protein interaction (PPI) network analyses of comparison-specific (61 and 25) and comparison-overlapping (78 and 22) regulated proteins. Bars and protein nodes are stained according to the colors displayed in the Venn diagram that represent overlapping and nonoverlapping subgroup comparisons.

**Figure 2 cancers-16-00008-f002:**
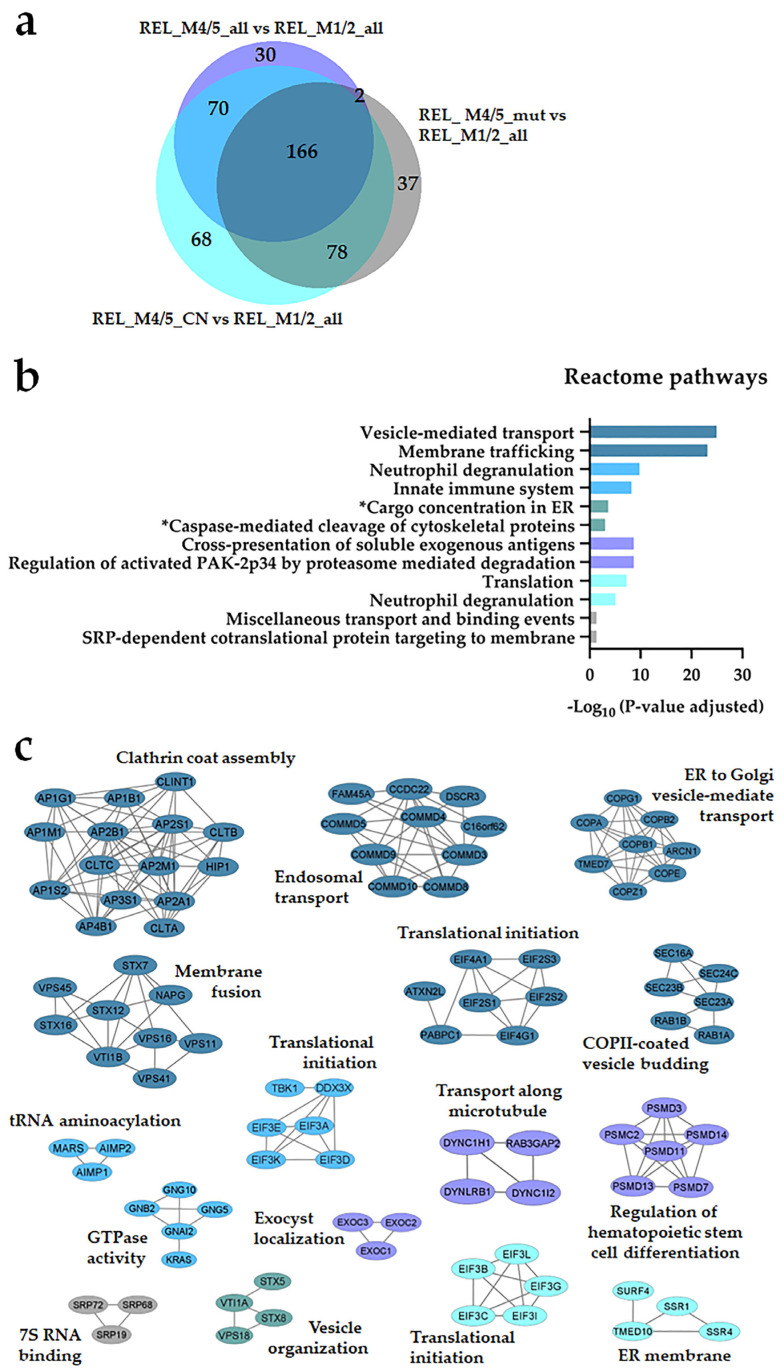
The AML cell proteome shows increased translation, neutrophil degranulation, and intracellular protein/vesicle transport proteins for RELAPSE patients with FAB classification M4 and M5. (**a**) Venn diagram of regulated proteins obtained from REL_M4/5_all vs. REL_M1/2_all, REL_M4/5_mut vs. REL_M1/2_all, and REL_M4/5_CN vs. REL_M1/2_all comparisons. (**b**) Reactome pathways and (**c**) PPI network analyses of comparison-specific (30, 37, and 68) and comparison-overlapping (70, 166, and 78) regulated proteins. Bars and protein nodes are stained according to the colors displayed in the Venn diagram that represent overlapping and nonoverlapping subgroup comparisons. * It stands for Reactome pathways with unadjusted *p*-value < 0.05.

**Figure 3 cancers-16-00008-f003:**
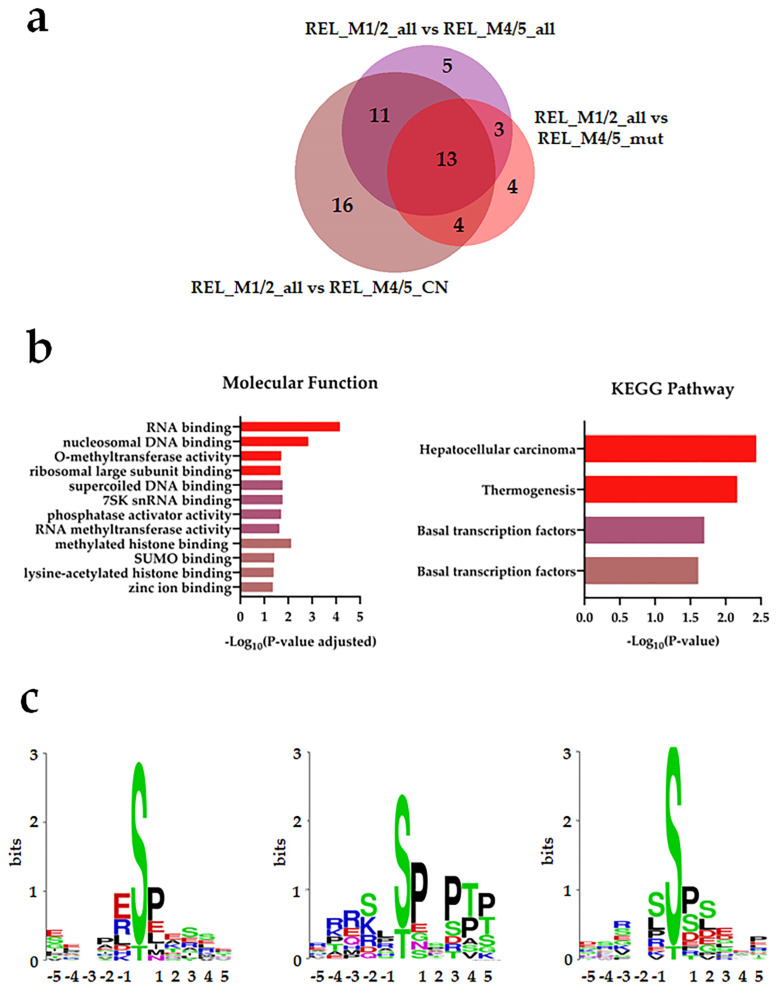
The AML cell phosphoproteome shows increased phosphorylation of RNA and DNA binding proteins for RELAPSE patients with FAB classification M1 and M2. (**a**) Venn diagram of differentially regulated phosphorylation sites (ANOVA, post hoc Turkey’s HSD with FDR < 0.05) obtained from REL_M1/2_all vs. REL_M4/5_all, REL_M1/2_all vs. REL_M4/5_mut, and REL_M1/2_all vs. REL_M4/5_CN comparisons. (**b**) Gene ontology (GO) with molecular function terms and KEGG pathway analyses of comparison-specific (16) and comparison-overlapping (11 and 13) differentially phosphorylated proteins. Bars are stained according to the colors displayed in the Venn diagram that represent overlapping and nonoverlapping subgroup comparisons. (**c**) Sequence motif analysis of the ±5 amino acids flanking the differentially regulated phosphorylation sites from the comparison-specific (16) on the right and comparison-overlapping (11 and 13) datasets in the middle and on the left, respectively.

**Figure 4 cancers-16-00008-f004:**
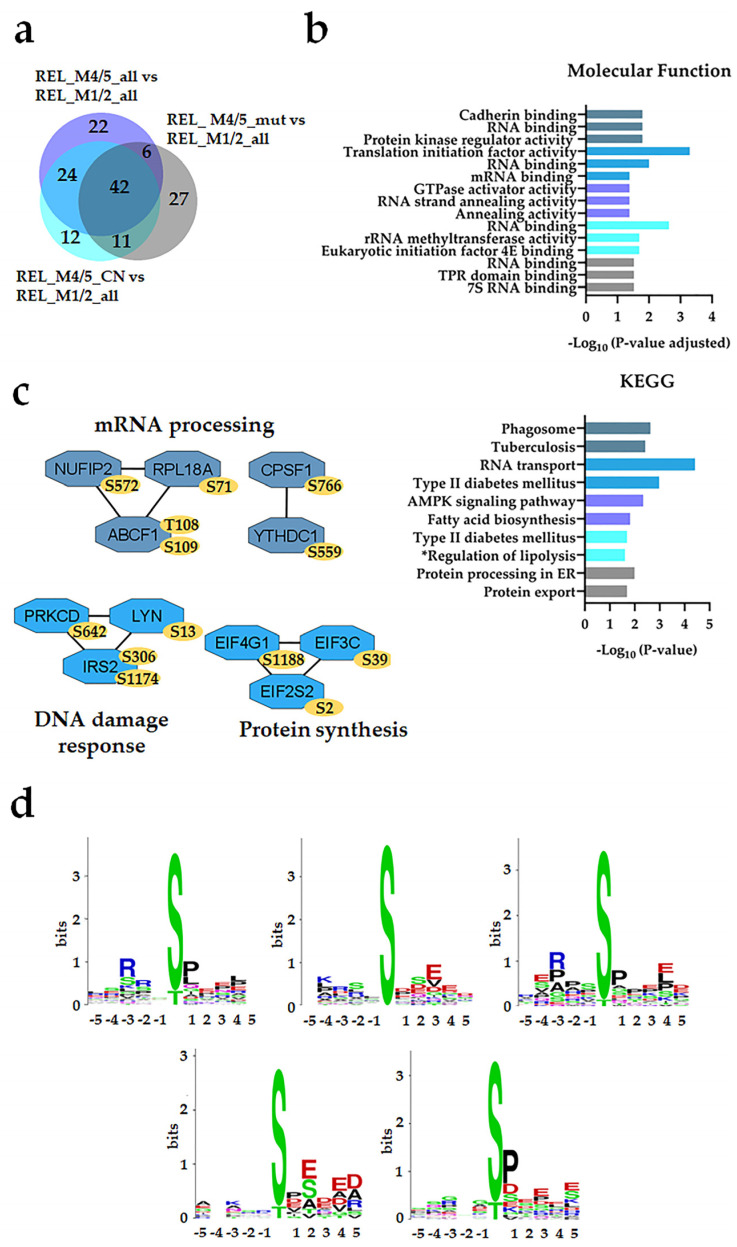
The AML cell phosphoproteome shows increased phosphorylation of RNA binding and transcriptional proteins for RELAPSE patients with FAB classification M4 and M5. (**a**) Venn diagram of differentially regulated phosphorylation sites obtained from REL_M4/5_all vs. REL_M1/2_all, REL_M4/5_mut vs. REL_M1/2_all, and REL_M4/5_CN vs. REL_M1/2_all comparisons. (**b**) GO with molecular function terms and KEGG pathway analyses of comparison-specific (22, 12, and 27) and comparison-overlapping (42 and 24) differentially phosphorylated proteins. (**c**) PPI network analyses of overlapping (42 and 24) differentially phosphorylated proteins. Bars and protein nodes are stained according to the colors displayed in the Venn diagram that represent overlapping and nonoverlapping subgroup comparisons. (**d**) Sequence motif analysis of the ±5 amino acids flanking the differentially regulated phosphorylation sites from the comparison-specific (22, 12, and 27) on top, right, on bottom, left and bottom, right, respectively, and comparison-overlapping (42 and 24) datasets on top, left and on top, middle, respectively. * A shorter name for “Regulation of lipolysis in adipocytes” KEGG pathway is added for space purposes.

**Figure 5 cancers-16-00008-f005:**
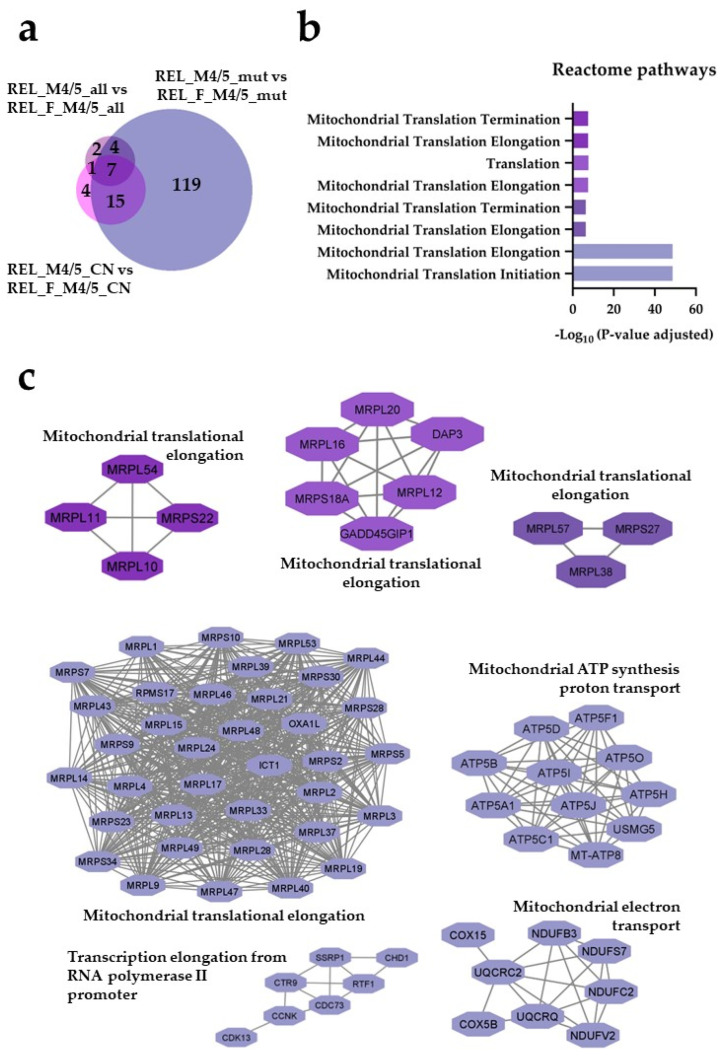
The AML cell proteome shows increased abundance of mitochondrial translational proteins for RELAPSE patients when compared with REL_FREE with FAB classification M4 and M5. (**a**) Venn diagram of regulated proteins obtained from REL_M4/5_all vs. REL_F_M4/5_all, REL_M4/5_mut vs. REL_F_M4/5_mut, and REL_M4/5_CN vs. REL_F_M4/5_CN comparisons. (**b**) Reactome pathway and (**c**) PPI network analyses of comparison-specific (119) and comparison-overlapping (15, 7, and 4) regulated proteins. Bars and protein nodes are stained according to the colors displayed in the Venn diagram that represent overlapping and nonoverlapping subgroup comparisons.

**Figure 6 cancers-16-00008-f006:**
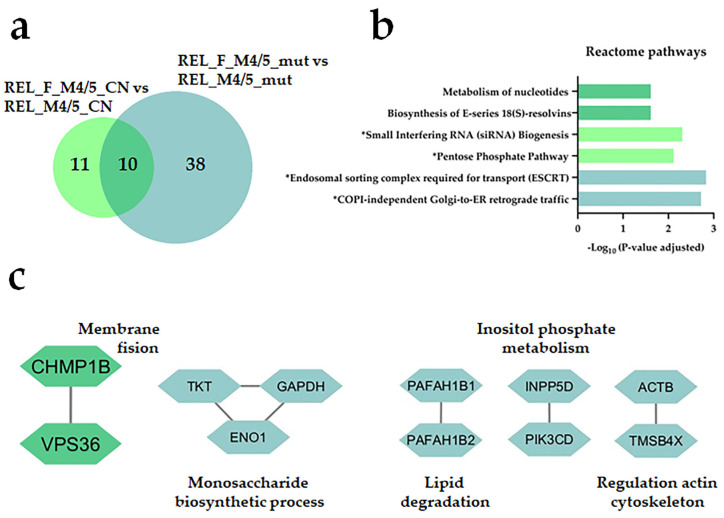
The AML cell proteome shows increased of proteins involved in the metabolism of nucleotides for REL_FREE patients when compared with RELAPSE with FAB classification M4 and M5. (**a**) Venn diagram of regulated proteins obtained from REL_F_M4/5_mut vs. REL_M4/5_mut, and REL_F_M4/5_CN vs. REL_M4/5_CN comparisons. (**b**) Reactome pathway and (**c**) PPI network analyses of comparison-specific (38 and 11) and comparison-overlapping (10) regulated proteins. Bars and protein nodes are stained according to the colors displayed in the Venn diagram that represent overlapping and nonoverlapping subgroup comparisons. * It stands for Reactome pathways with unadjusted *p*-value < 0.05.

**Figure 7 cancers-16-00008-f007:**
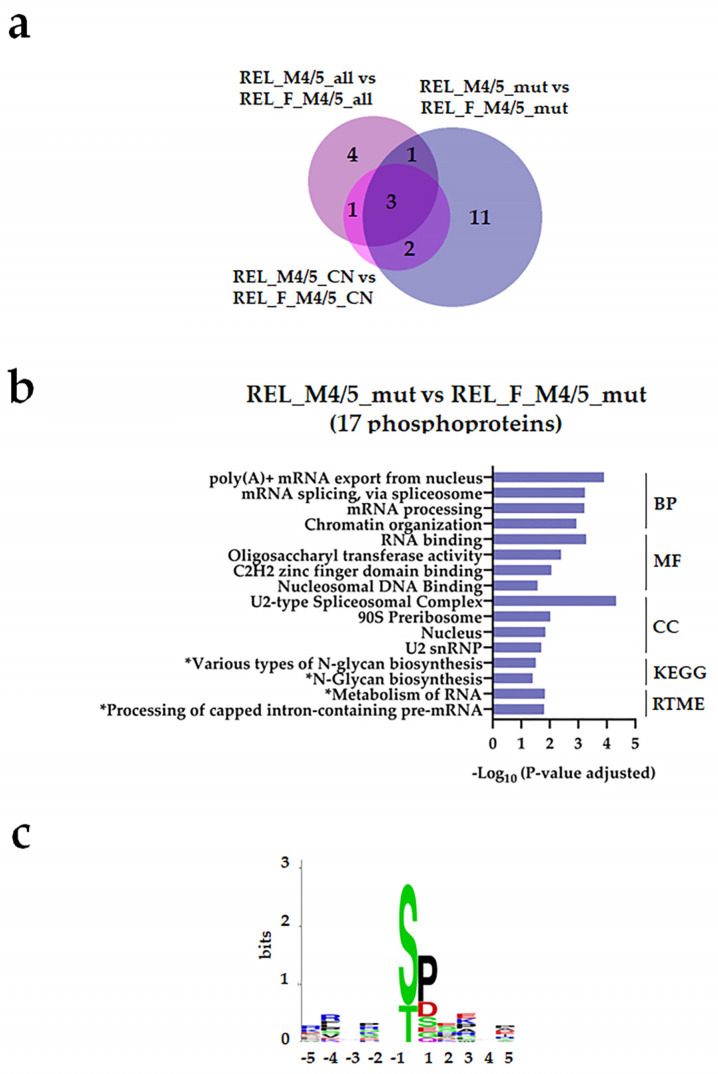
The AML cell phosphoproteome shows increased phosphorylation of RNA and DNA binding proteins for RELAPSE patients with FAB classification M4 and M5 and with the *NPM1* Ins mutation. (**a**) Venn diagram of differentially regulated phosphorylation sites obtained from REL_M4/5_all vs. REL_F_M4/5_all, REL_M4/5_mut vs. REL_F_M4/5_mut, and REL_M4/5_CN vs. REL_F_M4/5_CN comparisons. (**b**) GO with biological process (BP), molecular function (MF), and cellular compartment (CC) terms, KEGG and Reactome pathway analyses of 17 differentially higher phosphorylated proteins in the REL_M4/5_mut vs. REL_F_M4/5_mut comparison. (**c**) Sequence motif analysis of the ±5 amino acids flanking the differentially regulated phosphorylation sites in the REL_M4/5_mut vs. REL_F_M4/5_mut comparison. * It stands for KEGG and Reactome pathways with unadjusted *p*-value < 0.05.

**Figure 8 cancers-16-00008-f008:**
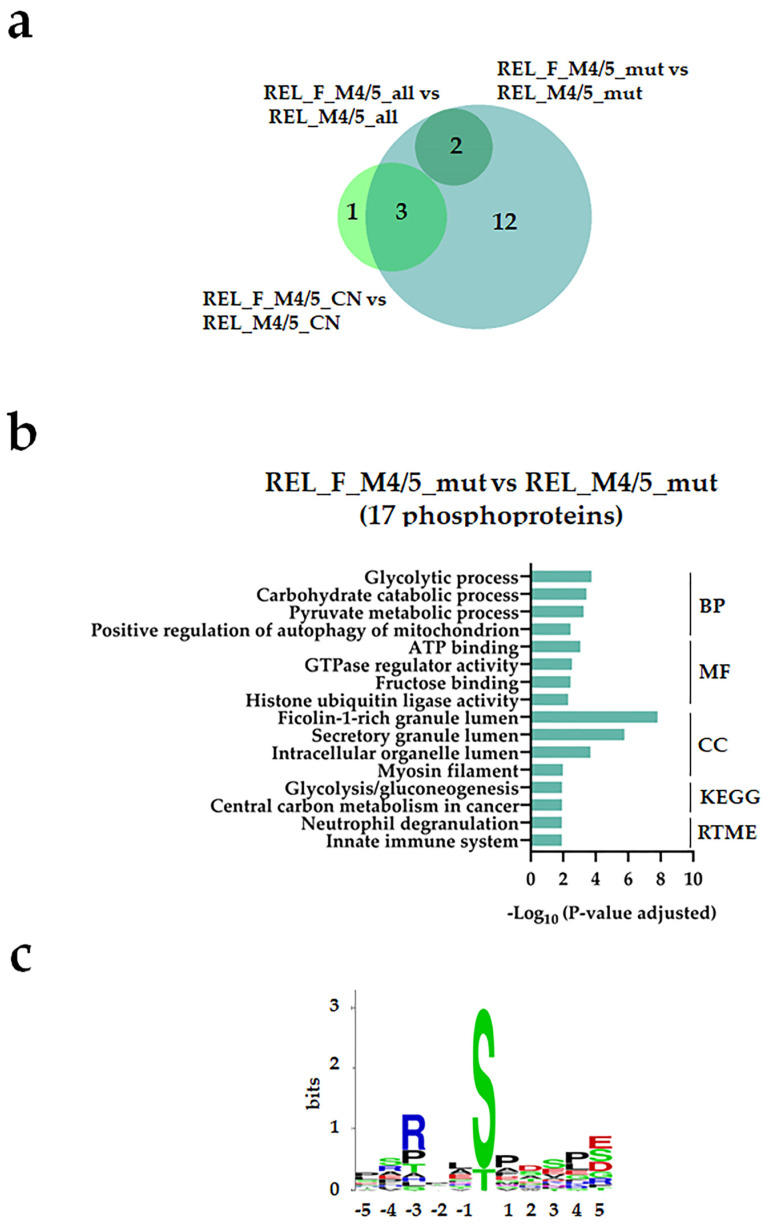
The AML cell phosphoproteome shows increased phosphorylation of glycolysis and gluconeogenesis proteins for RELAPSE_FREE patients with FAB classification M4 and M5 and with the *NPM1* Ins mutation. (**a**) Venn diagram of differentially regulated phosphorylation sites obtained from REL_F_M4/5_all vs. REL_M4/5_all, REL_F_M4/5_mut vs. REL_M4/5_mut, and REL_F_M4/5_CN vs. REL_M4/5_CN comparisons. (**b**) GO with BP, MF, and CC terms, KEGG and Reactome pathway analyses of 17 differentially higher phosphorylated proteins in the REL_F_M4/5_mut vs. REL_M4/5_mut comparison. (**c**) Sequence motif analysis of the ±5 amino acids flanking the differentially regulated phosphorylation sites in the REL_F_M4/5_mut subgroup when compared to REL_M4/5_mut patients.

**Table 1 cancers-16-00008-t001:** Characteristics of acute myeloid leukemia (AML) disease subgroups.

Characteristic		RELAPSE	REL_FREE
**FAB classification**	M1/M2	8	1
	M4/M5	12	14
** *NPM1* **	WT	6	6
	Ins	5	8
**CN**	46, XY or XX	7	9

FAB: French-American-British; *NPM1*: nucleophosmin 1; CN: cytogenetically normal; WT: wild type; Ins a 4 bp-insertion.

## Data Availability

All raw data and MaxQuant output files from the original cohort can be found in the ProteomeXchange consortium with the dataset identifier PXD014997.

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
