# Peer review of "High Mitochondrial Protein Expression as a Potential Predictor of Relapse Risk in Acute Myeloid Leukemia Patients with the Monocytic FAB Subtypes M4 and M5"

_cancers, 2023, doi:10.3390/cancers16010008_

Round 1

Reviewer 1 Report

Comments and Suggestions for Authors

In this proteomics study, a significant association between high mitochondrial protein abundance, cellular respiration, and clinical outcomes in Acute Myeloid Leukemia (AML) was found. Notably, elevated levels correlate with higher relapse rates and reduced overall survival. The identification of subtype-specific protein expression profiles, as exemplified in FAB subtypes M1/M2 and M4/M5, holds promise for discovering valuable biomarkers. Particularly, heightened expression of mitochondrial ribosomal proteins and associated respiratory chain complexes emerges as a reliable predictor of high relapse risk in AML patients with M4 and M5 subtypes. These findings offer potential targets for personalized therapies and underscore the importance of integrating proteomics data for improved prognostication in AML.

In considering recent mitochondrial proteomics studies in AML, it's evident that while the traditional FAB classification offers insights, it's somewhat outdated in the current genomic era. The focus on FAB subtypes M1/M2 and M4/M5 reveals a connection between mitochondrial factors and clinical outcomes. One of the study's limitation lies also in the relatively small patient sample size.

However, the study appears to be technically careful, detailed and precise in its methodology. Despite the discussed limitations, these findings merit consideration for publication, contributing valuable insights to the evolving landscape of AML research.

Author Response

REVIEWER 1

In this proteomics study, a significant association between high mitochondrial protein abundance, cellular respiration, and clinical outcomes in Acute Myeloid Leukemia (AML) was found. Notably, elevated levels correlate with higher relapse rates and reduced overall survival. The identification of subtype-specific protein expression profiles, as exemplified in FAB subtypes M1/M2 and M4/M5, holds promise for discovering valuable biomarkers. Particularly, heightened expression of mitochondrial ribosomal proteins and associated respiratory chain complexes emerges as a reliable predictor of high relapse risk in AML patients with M4 and M5 subtypes. These findings offer potential targets for personalized therapies and underscore the importance of integrating proteomics data for improved prognostication in AML.

1.1. In considering recent mitochondrial proteomics studies in AML, it's evident that while the traditional FAB classification offers insights, it's somewhat outdated in the current genomic era. The focus on FAB subtypes M1/M2 and M4/M5 reveals a connection between mitochondrial factors and clinical outcomes. One of the study's limitations lies also in the relatively small patient sample size.

Response: The differentiation block is a fundamental characteristic of AML cells, but despite this we agree that a focus on differentiation seems outdated in the current genomic era that is also reflected in the most recent WHO 2022 classification. However, based on several recent studies the degree of AML cell differentiation seems relevant for the future era of targeted AML therapy. This is discussed more in detail in a new chapter of the Discussion section. In our opinion the more detailed characterization (including proteomic/phosphoprotein profiling) is relevant for a better understanding of the differentiation block and the biological importance of differentiation induction in human AML.

We have included an additional comment on differentiation and targeted therapy at the beginning of the second chapter of the Discussion section. This new chapter illustrates the importance of differentiation/differentiation induction for several forms of new targeted AML therapies.

Furthermore, we would also emphasize that we use the FAB classification because it represents a standardized and accepted tool for classification of patients with regard to degree of differentiation block, and the detailed information for each individual patient. New Table S1 and Table S2 give all our available information that is relevant for classification of patients according to the WHO 2022 classification (previous chemotherapy/hematological disease, karyotype and molecular genetic analyses; see also our response to comment 3.1).

These new data tables are commented in a new Section 3.1. The patient number is also commented in the last chapter of this new Section 3.1. 

1.2. However, the study appears to be technically careful, detailed and precise in its methodology. Despite the discussed limitations, these findings merit consideration for publication, contributing valuable insights to the evolving landscape of AML research.

Response: We are grateful for this general comment.

Reviewer 2 Report

Comments and Suggestions for Authors

Dear authors, while this work is of interest and could point to mechanisms of relapse in different FAB AML cohorts, I have some questions.

In the abstract you refer to your patient cohort as being chemoresistant/relapse - is there a mixture of refractory and relapse patients? If so it would be interesting to separate these into two subgroups.

From your data, would it not have been more beneficial to compare relapse to relapse_free and then the differences associated with the relapse cohorts taken forward for comparisons between M1/2 and M4/5. Therefore only exploring/comparing aberrations associated with relapse. 

Was the comparison between relapse and relapse free for M1/2 subgroups not performed? Or did it not show anything interesting?

While this work is somewhat interesting the observations made have been drawn from a small patient cohort - it would be nice to try and strengthen these findings in another dataset. Perhaps the TARGET data - whilst data from children - you do draw a conclusion in the discussion about FAB classification and relapse/death following HSCT in children.

Perhaps include the age range of your patient cohort (I know it was published in your previous manuscript, but it would be nice for readers of this article to know the disease being investigated). 

Author Response

REVIEWER 2

Dear authors, while this work is of interest and could point to mechanisms of relapse in different FAB AML cohorts, I have some questions.

2.1. In the abstract you refer to your patient cohort as being chemoresistant/relapse - is there a mixture of refractory and relapse patients? If so it would be interesting to separate these into two subgroups.

Response: We apologize for this misunderstanding. All our patients achieved complete hematological remission after intensive induction chemotherapy, and a subset of the patients later developed chemoresistant relapse. We have rewritten this part of the abstract to avoid misunderstandings.

2.2. From your data, would it not have been more beneficial to compare relapse to relapse_free and then the differences associated with the relapse cohorts taken forward for comparisons between M1/2 and M4/5. Therefore only exploring/comparing aberrations associated with relapse. 

Response: This strategy could not be used due to the low number of FAB-M1/M2 patients in our study (see Table 1). Please see the comment in the last chapter of the new Section 3.1.

2.3. Was the comparison between relapse and relapse free for M1/2 subgroups not performed? Or did it not show anything interesting?

Response: These two groups were too small to allow a statistical comparison (see Table 1). Please see the comment …Due to the low number of patients, FAB-M1/M2 long-term survivors are not included in this study..” in the new Section 3.1.

2.4. While this work is somewhat interesting the observations made have been drawn from a small patient cohort - it would be nice to try and strengthen these findings in another dataset. Perhaps the TARGET data - whilst data from children - you do draw a conclusion in the discussion about FAB classification and relapse/death following HSCT in children.

Response: The following information is given about the TARGET dataset on their own website:  

There are 130 fully characterized patient cases that make up the TARGET AML dataset, each with gene expression, tumor and paired normal copy number analyses, and at least one type of sequencing data available. This cohort includes 50 patients with adequate relapse specimens to study as trios. There are additional cases with partial molecular characterization and/or sequencing data available, which can be sorted via the Case Matrix. dbGaP Study (nih.gov)

The TARGET Acute Myeloid Leukemia projects employed comprehensive molecular characterization to determine the genetic changes that drive the initiation and progression of high-risk or hard-to-treat childhood cancers. Acute myeloid leukemia (AML) is a cancer that originates in the bone marrow from immature white blood cells known as myeloblasts. About 25% of all children with leukemia have AML. Although survival rates have increased since the 1970s, approximately half of all childhood AML cases relapse despite intensive treatment. Additional therapies following relapse are often unsuccessful and can be especially difficult and damaging for children. These patients would clearly benefit from targeted therapeutic approaches.

Through comprehensive genome-wide characterization, TARGET researchers are identifying the genetic and epigenetic alterations of relapsed disease. The ultimate goal is to translate their discoveries into novel treatments that will improve outcomes for children with AML. To learn more about pediatric AML and current treatment strategies, visit the NCI pediatric AML website.

TARGET AML molecular characterization analyses include gene expression array, copy number array, DNA methylation, Whole Genome Sequencing, Whole Exome Sequencing, RNA-seq, miRNA-seq and Targeted Capture Sequencing. TARGET Acute Myeloid Leukemia (AML) | NCI Genomic Data Commons (cancer.gov)

- Due to the biological importance of posttranscriptional regulation of individual protein levels it will in our opinion be difficult to use datasets based on gene expression analyses to verify our observations at the proteomic/phosphoproteomic levels. For this reason, we have not used the TARGET dataset for verification. To the best of our knowledge no proteomic/phosphoproteomic dataset is available for verification. We hope this argument can be considered as relevant by the reviewer.

2.5. Perhaps include the age range of your patient cohort (I know it was published in your previous manuscript, but it would be nice for readers of this article to know the disease being investigated). 

Response: The age of the individual patients can be seen from the new Table S1 and Table S2.

Reviewer 3 Report

Comments and Suggestions for Authors

The patients are not stratified according to current WHO classification. The authors should respond to that or explain (and describe as limitation of the study).

Was any of the relapsed patients treated with BH3-mimetics?

It would be great do deconvolute BCL2-family proteins from the proteomic data. Are they differentially expressed?

Comments on the Quality of English Language

No major problems.

Author Response

REVIEWER 3

3.1. The patients are not stratified according to current WHO classification. The authors should respond to that or explain (and describe as limitation of the study).

Response: We have now included more detailed patient information in new supplementary Table S1 and Table S2 that has also been reorganized compared with the differentiation-based classification in our previous study. We present and comment our patient subclassification that is based on the AML cell differentiation block, in the context of the most recent WHO 2022 and ELN 2022 classification in a new Section 3.1.

3.2. Was any of the relapsed patients treated with BH3-mimetics?

Response: The BH3 mimetics are small molecule antagonists of the anti-apoptotic BCL-2 family members that function as competitive inhibitors by binding to the hydrophobic cleft. None of the patients received such targeted therapy; this is now stated in Section 3.1,

3.3. It would be great do deconvolute BCL2-family proteins from the proteomic data. Are they differentially expressed?

Response: We are grateful for this suggestion. We have now included analysis of our patients based on the expression level of all identified proapoptotic and antiapoptotic BCL-2 family members. We found a high abundance of Bid in non-relapsing AML patients with the FAB subtypes M4/M5. This is presented in a new chapter in the Results section (chapter 3.4 and briefly commented on in a new chapter in the Discussion section (page 20)).

Round 2

Reviewer 2 Report

Comments and Suggestions for Authors

Dear authors,

Thank you for making the relevant changes and explaining the ones you could not. Based on this I am satisfied that the manuscript can be published. 

Reviewer 3 Report

Comments and Suggestions for Authors

The manuscript is now improved and publication is justified.

Comments on the Quality of English Language

English is fine.